# Remote Sensing Based Binary Classification of Maize. Dealing with Residual Autocorrelation in Sparse Sample Situations

**Mario Gilcher [1],\*, Thorsten Ruf [2], Christoph Emmerling [2] and Thomas Udelhoven [1]**

[1] Department of Remote Sensing and Geoinformatics, Faculty of Regional and Environmental Sciences, University of Trier, Campus II, D-54286 Trier, Germany; udelhoven@uni-trier.de

[2] Department of Soil Science, Faculty of Regional and Environmental Sciences, University of Trier, Campus II, D-54286 Trier, Germany; ruf@uni-trier.de (T.R.); emmerling@uni-trier.de (C.E.)

\* Correspondence: gilcher@uni-trier.de; Tel.: +49-(0)-651-201-4607

**Abstract:** In order to discuss potential sustainability issues of expanding silage maize cultivation in Rhineland-Palatinate, spatially explicit monitoring is necessary. Publicly available statistical records are often not a sufficient basis for extensive research, especially on soil health, where risk factors like erosion and compaction depend on variables that are specific to every site, and hard to generalize for larger administrative aggregates. The focus of this study is to apply established classification algorithms to estimate maize abundance for each independent pixel, while at the same time accounting for their spatial relationship. Therefore, two ways to incorporate spatial autocorrelation of neighboring pixels are combined with three different classification models. The performance of each of these modeling approaches is analyzed and discussed. Finally, one prediction approach is applied to the imagery, and the overall predicted acreage is compared to publicly available data. We were able to show that Support Vector Machine (SVM) classification and Random Forests (RF) were able to distinguish maize pixels reliably, with kappa values well above 0.9 in most cases. The Generalized Linear Model (GLM) performed substantially worse. Furthermore, Regression Kriging (RK) as an approach to integrate spatial autocorrelation into the prediction model is not suitable in use cases with millions of sparsely clustered training pixels. Gaussian Blur is able to improve predictions slightly in these cases, but it is possible that this is only because it smoothes out impurities of the reference data. The overall prediction with RF classification combined with Gaussian Blur performed well, with out of bag error rates of 0.5% in 2009 and 1.3% in 2016. Despite the low error rates, there is a discrepancy between the predicted acreage and the official records, which is 20% in 2009 and 27% in 2016.

**Keywords:** crop classification; spatial autocorrelation; Regression Kriging

## 1. Introduction

### 1.1. Motivation

Expanded silage maize cultivation, as a direct consequence of Germany's energy turnaround and more precisely the governmentally guaranteed feed-in-tariffs [1,2], comes with several potentially negative externalities. The high management intensity of silage maize and the elevated threat of soils towards erosion, observed losses of agrobiodiversity as well as the trend towards monotonous landscape structures are calling the sustainability of such cultivation systems into question [3]. Consequently, spatially explicit monitoring is necessary, which may facilitate a risk assessment towards

erosion rates. Defining critical limits for soil erosion may help to make silage maize cultivation subject to conditions of erosion protection methods to distinctly reduce the risk of soil loss or exclusion of highly vulnerable sites. Moreover, such analysis may be used to support political decisions. Although these kinds of datasets are tracked by administrative bodies on the state level, they are rarely available to researchers. Thus, land-use classification approaches using optical remote sensing imagery could allow for a more independent and conclusive research. While the paper by Ruf [4] focuses more on the application and the implications of such an analysis, this paper addresses the major methodological obstacles from this study, which are very specific for spatially explicit pixel based classification of satellite imagery.

### 1.2. Monitoring Land Use Dynamics with Pixel Based Classification of Optical Remote Sensing Imagery

Land cover classification is one of the most popular applications for remote sensing scientists. Yu et al. [5] analyzed the popularity of land cover classification algorithms, and assessed that the majority of publications used the Maximum Likelihood (ML) classification, which is comparatively simple and widely implemented in remote sensing software packages. At the same time, more advanced classification algorithms like Support Vector Machines (SVM) and ensemble based methods perform land cover classification considerably better. A meta study analyzing the performance of supervised pixel-based classification algorithms specifically focusing on studies where performances of algorithms were compared to compute a quantitative synthesis of 266 articles [6]. It specifically identified SVMs, Neural Networks (NN) and Random Forest (RF) classifiers as superior to statistical tools like ML classification. However, the median differences of Overall Accuracy (OA) between SVM, NN and RF were all less than 2% or equal, which means that these rather modern machine-learning algorithms performed comparably in most cases.

In contrast to a complete landcover classification of a certain region, in this study, only the detection of maize fields was of importance. This means that only one class exists on a conceptual level, and therefore only one type of training data. One approach to deal with this problem is to adjust the classification algorithm. For instance, several adjusted versions exist of the SVM algorithm, which was initially conceived as a binary classifier [7]. Another approach is to sample from the unlabeled observations, to get two sets of training data, which are either positive or unlabeled, which is why this approach is called PU learning [8]. The downside here is that the unlabeled observations can contain a varying amount of positive pixels. Since the reference data are generated based on orthophotos (see Section 2.3), the distinction of maize and other pixels can be made quite clearly; therefore, two classes can be produced. One of these classes is very homogeneous (maize), the other is very heterogeneous and can contain pixels like pastures which are very close to maize, and bare soil which is very different from maize.

The intended pixel based classification of the multispectral imagery misspecifies the problem insofar, as it treats each pixel independently. Every observation is a numeric vector, consisting of a number of reflectance values from the satellite imagery, and a categorical dependent variable that indicates the presence of maize at that particular location. The location itself, along with the implications of proximity and neighborhood to other pixels, is not included. Since maize fields are on average between 0.5 and 2 hectares, which is much larger than a single pixel, the presence of maize in one pixel increases the probability of maize for all pixels in close proximity. This phenomenon is called spatial autocorrelation [9].

There are many different angles from which this issue can be approached [10], beginning with the sample selection [11] and sample refinement [12], via the classification algorithm itself [13] and ending with adjustments made post-classification [14]. In spite of that, spatial relationships are still rarely analyzed in remote sensing classification research. The aforementioned review papers on classification of remote sensing imagery ([5,6]) do not cover the topic at all. Another recent review paper [15] specifically on RF based classifiers only mentions the issue of spatially autocorrelated training data.

Therefore, while many different ways exist to incorporate spatial relationships of pixels, their usage is not widespread and most applications are still aspatial.

### 1.3. Objectives

This study aimed to establish a validated toolchain to prove the suitability of optical remote sensing as a primary source to monitor the prevalence of maize in a given region, which has potential sustainability implications for medium scale regions. For this purpose, well understood and performant classification algorithms were employed in conjunction with two different methods to incorporate pixel relationships with their immediate neighborhood. This includes the following objectives:

1. **Systematic assembly of reference data for the study area**:
   To produce uniformly distributed reference data, aerial images from 2009 and 2016 were used to digitize maize and non-maize polygons for the entire study region. For both years, two kinds of reference datasets have been produced. A sparse dataset, with a minimum of six polygons per 25 km$^2$, covers the entire study area. Additionally, five dense datasets for 1.5 by 1.5 km subsets were produced, where all fields were digitized manually.
2. **The preprocessing of remote sensing imagery**:
   In the given years, two RapidEye scenes from late August were available. The tiles were preprocessed, in order to get one multispectral observation for each pixel of the study area.
3. **Small scale model evaluation**:
   The dense reference dataset for the five subsets was then used to analyze and optimize model behavior on a small scale. Three classification techniques (Generalized Linear Model, Random Forests, Support Vector Machines) were combined with two methods to account for spatial autocorrelation (Simple Kriging, Gaussian Blur).
4. **Medium scale model application for the entire dataset**:
   The optimized modeling approach was then trained with the sparse medium scale reference dataset, and applied to the entire region. Performance was analyzed and results were aggregated and visualized.

## 2. Materials and Methods

### 2.1. Study Area

The study was conducted in the "Eifelkreis Bitburg-Prüm" (in the following referred to as 'study area'), which is an administrative district in the southwest of Germany, bordering Luxembourg and Belgium. It has a size of 1627 km$^2$ and mostly consists of low mountain ranges, with a slow gradient in elevation from roughly 300 masl. in the south up to around 700 masl. in the northern part of the study area. Its population is sparsely distributed, and the rural landscape mostly consists of a considerable share of forests, while 53.2% of the total area is used by agriculture (Statistisches Landesamt Rheinland-Pfalz [16]).The agriculture of the region is mostly pastureland, with some cereals, and a substantial increase in silage maize cultivation during the past 20 years. This increase coincides with the progressive installation of biogas plants, starting with a first plant built in 2000, to 57 plants with a total installed capacity of 19,152 kW in the year 2016 [17]. As a result, the study area now shows the highest capacity of agricultural biogas producing units in Rhineland-Palatinate. The comparatively homogeneous rural structure in conjunction with the clear regional shift in cultivation strategies as a direct consequence of policymaking offers a unique opportunity to illustrate the capabilities of optical remote sensing in detecting land cover changes.

### 2.2. Image Data

The selection of image datasets was constrained by requirements of resolution, availability, phenology window, cloud coverage and time frame, since reference data was only available in 2009

and 2016 (see Section 2.3). The spatial resolution has to be adequate, with many fields smaller than a few hectares. The RapidEye Science Archive [18] offers a unique opportunity to researchers by providing multispectral imagery for research projects. RapidEye is a constellation of five satellites in a 630 km sun-synchronous orbit, with a nadir Ground Sampling Distance of 6.5 m, which is orthorectified to a pixel size of 5 m. The sensor has three bands in the visible range, one red edge band, as well as one band in the near infrared range [19]. The revisit time of approximately six days increases the chances to get imagery on cloud free days, while the general timing window of the phenologic cycle of maize is met. Of all the crops used in the study area, maize reaches the peak vegetation cover the latest, around the end of August. At that time, rape and cereals have typically been already harvested, and maize is largely the only agricultural crop still standing [20]. This leaves a timing window of about four weeks between mid-August until mid-September.

Within these constraints, two scenes in late August could be identified, in the years 2009 and 2016. They were entirely cloud free, with only negligible amounts of missing pixels in the far northwest in 2009. The imagery was provided as 25 by 25 km tiled Level 3A Geotiffs, which means they were orthorectified and radiometrically corrected. These tiles were then atmospherically corrected based on the 5S algorithm [21] with the software AtCPro (Version 6.0, Department of Remote Sensing and Geoinformatics, Trier, Germany) [22,23], mosaicked and cropped. In the atmospheric correction process, a digital elevation model (DEM) [24] was used to compensate for topographic effects. It was also used in the prediction models, since elevation has a significant influence on local climate and therefore phenology. After the atmospheric correction, the mosaic was masked in two steps. Firstly, this was to only include pixels within the borders of the study area (Nomenclature of Territorial Units for Statistics (NUTS)-3 Region Eifelkreis Bitburg Prüm). Secondly, this was to exclude roads, settlements and forests, i.e., only contain pixels in agricultural use (agricultural areas acc. to the German land appraisal, provided by the LGB-RLP).

*2.3. Reference Data*

Since the classification of around 87,000 hectares of agricultural land requires a large amount of evenly distributed training data, a visual survey approach was chosen, based on Google Earth aerial imagery (see Figure 1). The high resolution aerial images were used to identify maize fields, which was feasible because of the unique texture of grown maize fields. Their shape was then roughly digitized on top of the RapidEye images. At the same time, polygons were digitized with no maize fields inside. To guarantee an even distribution of training fields across the study site, it was split in subdivisions using a 9 by 12 grid, with each cell spanning 5 by 5 km. The main goal was to select a small set of no less than three polygons for both maize and non-maize land cover types separately. This was not always possible, especially since some cells around the fringes of the study area were mostly covered by forests. In addition, some cells in the east and west of the scene in 2016 were left empty because there was no high-resolution Google Earth imagery available for that date. Overall, around 2500 polygons were digitized for both years combined, amounting to over 1000 hectares of maize fields per year. This is a substantial percentage of the overall acreage, and should guarantee that the selected pixels were representative for the entire study area. In the following sections, this dataset is referred to as sparse medium scale reference.

To analyze spatial autocorrelation and use it to the classifiers' advantage, smaller subsets have been digitized independently from the grid based sparse reference dataset, but still based on the high-resolution imagery. For each of both years, five 1.5 km by 1.5 km squares were subsetted. They specifically included mostly pixels without settlements and very little roads, so that most pixels were covered either by maize, bare soil or vegetation different from maize. These ten square subsets were then digitized completely, aside from transitional pixels on the edges of fields and dirt roads. This lead to dense datasets with a moderate amount of overall observations, which were used to monitor spatial correlation effects especially for the model probability residuals. In the following sections, this dataset is referred to as dense small scale reference.

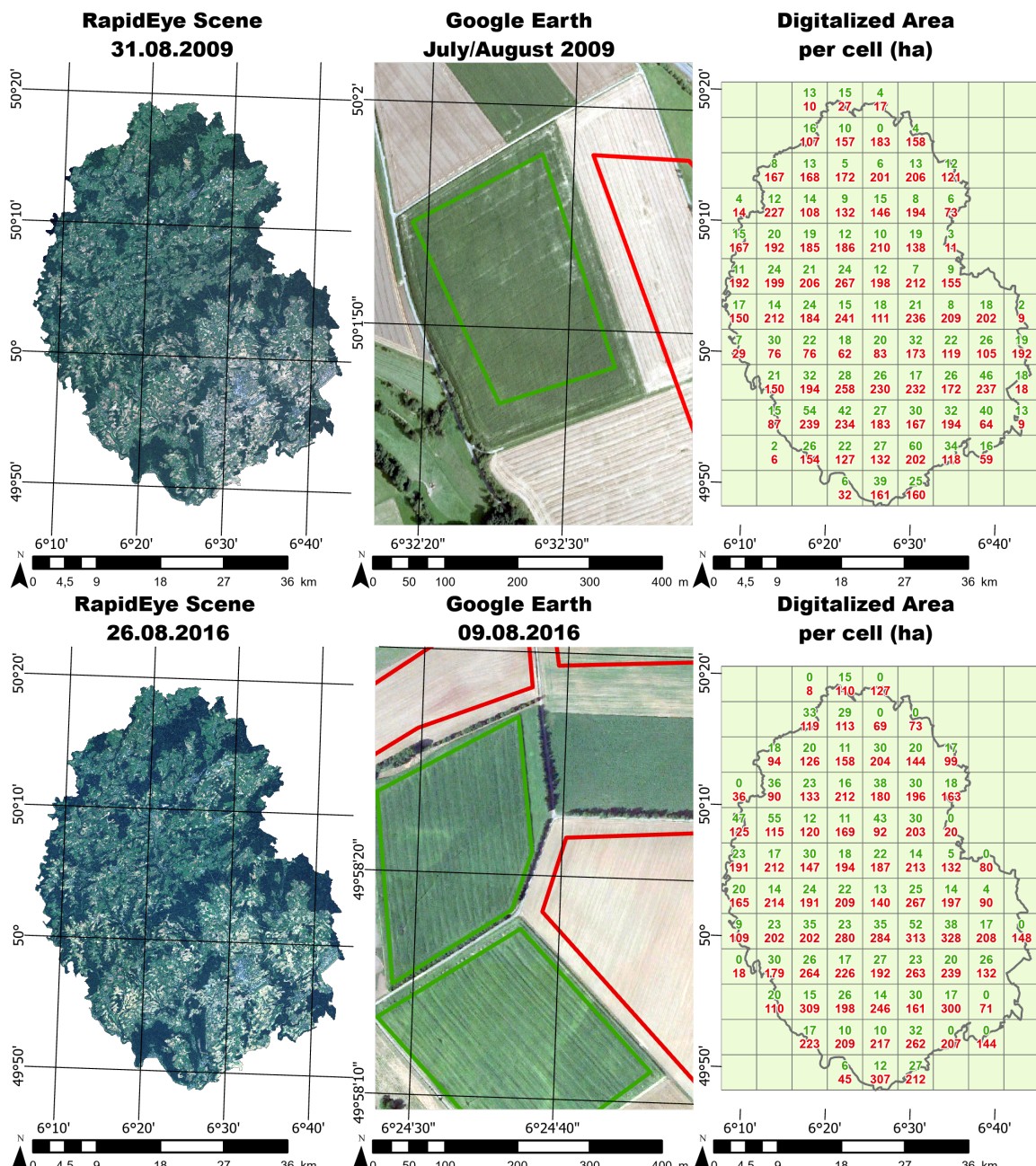

**Figure 1.** RapidEye coverage of the study area and manual digitization of reference data. The left column shows the two RapidEye scenes. The middle column shows the high resolution aerial imagery, and some digitized polygons. The green polygons are maize fields and the red polygons are a mixture of mostly bare soil and green lands. The right column shows the distribution of digitized area for each cell.

## 2.4. Modeling

### 2.4.1. Modeling Overview

The main goal of this study was to find ways to deal with spatial autocorrelation of residuals in the context of independent pixel based predictions. To achieve this, the modeling process was split up in two parts. The first part was the independent pixel based model, where a probability for a given pixel to be maize was estimated based on the covariates for this particular pixel, disregarding all

neighboring pixels. The second part took the spatial autocorrelation between neighboring pixels into account, and adjusted the probabilities for a given pixel accordingly.

### 2.4.2. Spatial Autocorrelation

Every observation is a numeric vector, consisting of five reflectance values from the satellite imagery, the altitude of the DEM, and a categorical dependent variable, which indicates the presence of maize at that particular location. The location itself, along with the implications of proximity and neighborhood to other pixels were not included. The Regression Kriging approach [25] allows for using spatial autocorrelation to the advantage of the classification process. By splitting each modelling problem into an independent pixel based modeling part (deterministic) and a spatial part (stochastic), many different pixel based modeling approaches can be combined with the Simple Kriging (SK) of their residuals, and therefore incorporate inherent spatial effects in an otherwise spatially independent modelling approach. The most common way to describe the relationship of probabilities being increasingly similar with proximity is the variogram. The empirical variogram describes the variance between pairs of observations with a fixed distance [26]. Figure 2 shows five different variograms from the small-scale digitized sites in 2016 (see Section 2.3). On all five sites, the variance increases steadily for the first few hundred meters, and plateaus at a distance between 300 and 500 m.

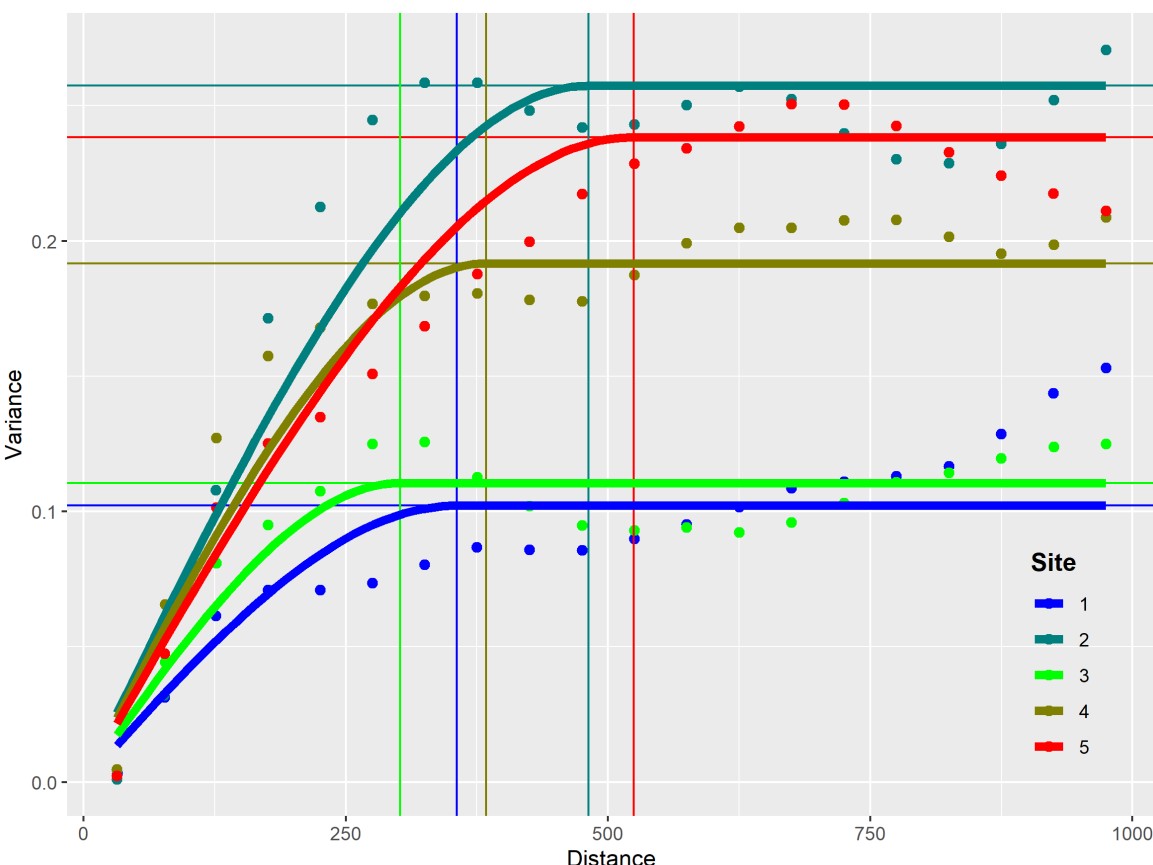

**Figure 2.** Variograms of five different sites in 2016. The points show the empirical variance of data points for the given distances. The bold lines show the fitted spherical models for each of the sites. The horizontal lines indicate the sill of the model, while the vertical lines indicate the range.

### 2.5. Research Design

The research design (see Figure 3) consisted mainly of three parts. The first part included all the preprocessing steps that were necessary to establish the datasets, to train, validate and apply the model. The results were the raster files with the predictors, a small scale reference dataset with dense

polygons for the five subsets (see Section 2.3), and a medium scale dataset with sparse polygons for the entire study area, each of which were available for both years.

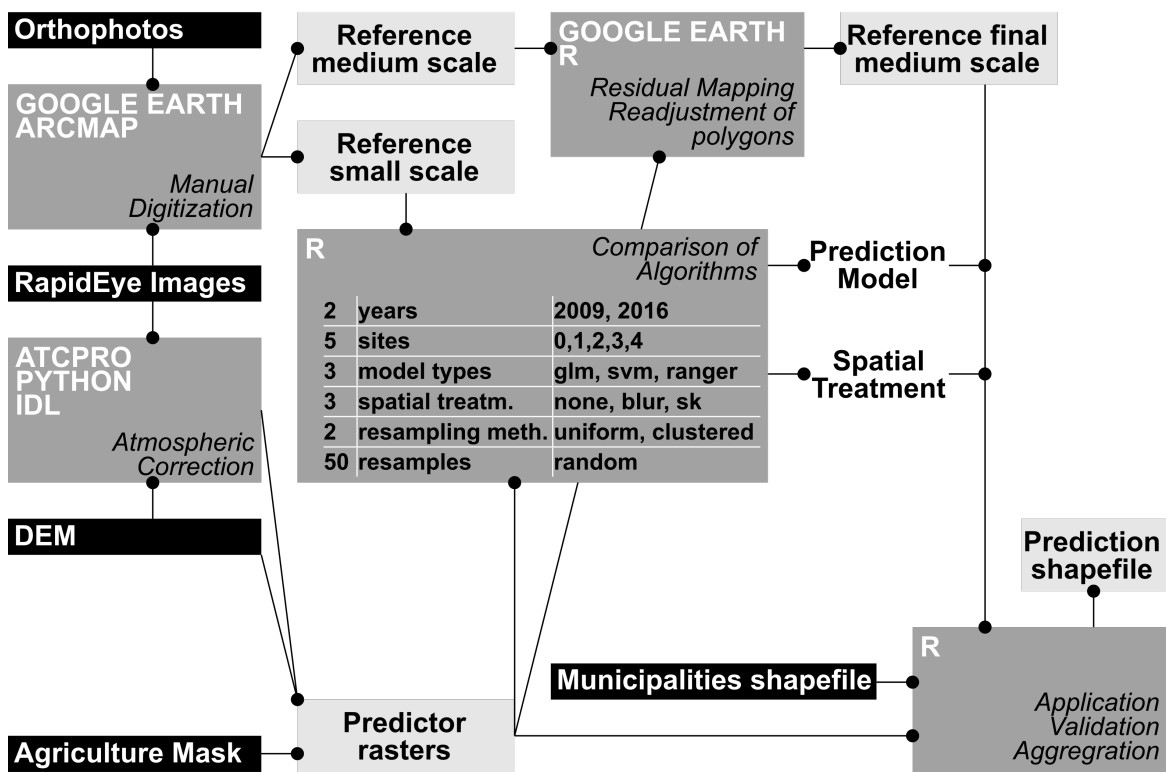

**Figure 3.** Research design. **1** : Preprocessing. The left side of the diagram shows the two major preprocessing steps of atmospheric correction and manual digitization, based on the given datasets shown in the black rectangles. **2**: Modelbuilding. The prediction model and spatial treatment were chosen with an extensive algorithm comparison based on the small scale reference data. The medium scale dataset was used to evaluate itself by mapping prediction residuals and adjusting critical areas. **3**: Application. The prediction model and the chosen spatial treatment was then applied to the final medium scale dataset, and the results were aggregated based on municipalities.

The second part of the research design, was the actual modeling. It was split into two distinct parts:

1. Small scale model evaluation
2. Visual evaluation and optimization of the medium-scale grid based sparse digitization of maize and non-maize polygons

In the first step, three different model types were compared, in combination with three (two + one control classification with independent pixels) different approaches to deal with spatial autocorrelation. The pixel based model types used were firstly the Generalized Linear Model (GLM) with binomial error distribution, also called Logistic Regression. Since the monitoring of maize occurrence is a binary classification problem, the GLM is a transparent and robust statistical approach that can then be compared to the two state-of-the-art machine-learning techniques of SVM and RF (also sometimes referred to as ranger, after the implementation algorithm in R). All three algorithms have the advantage that they are able to output probabilities, which is important to apply the techniques to reduce residual autocorrelation. Hyperparameters were not tuned at this stage because gridsearches would greatly inflate the runtime. Therefore, standard hyperparameters were used, with a radial SVM kernel. The classification probabilities predicted by these algorithms were then combined with three approaches of

dealing with residual autocorrelation. The first approach was a control treatment, focusing on the extent of the residual autocorrelation with just independent pixel based predictions. The second approach was the Gaussian blurring, where the probability of each pixel is a weighted sum of surrounding pixel based on a Gaussian kernel function. The third approach was Regression Kriging where the residuals of initially predicted probabilities are interpolated with Simple Kriging. The three classification models were chosen because they are popular, well understood and generally very performant, both in runtime and in accuracy. The RK approach was chosen to test one demanding geostatistical model (RK) which solves the problem of residual autocorrelation in an explicit way. It was compared to the very simple and runtime-economical Gaussian blurring.

Since clustering of training samples is a problem in Regression Kriging [25], another layer was added to the research design, as illustrated in Figure 4. The computation of these models, even in the context of these smaller subsamples is very time-consuming, which means that training samples have to be drawn randomly. For each individual run, 1000 sample pixels were drawn, 500 being maize, and 500 being not maize. The rest of the, on average, roughly 55,000, reference pixels were used to validate that particular model run. In order to truly test the applicability of the Regression Kriging approach, this clustering was simulated by randomly picking five polygons with and five polygons without maize to draw the random sample of 1000 training pixels from. This resampling was done 50 times for each combination of factors, which led to a total of 9000 model runs. For each run, performance indicators were computed to assess model performance. Additionally, the variogram objects were stored in order to visualize the spatial autocorrelation of residuals for each of the influencing factors.

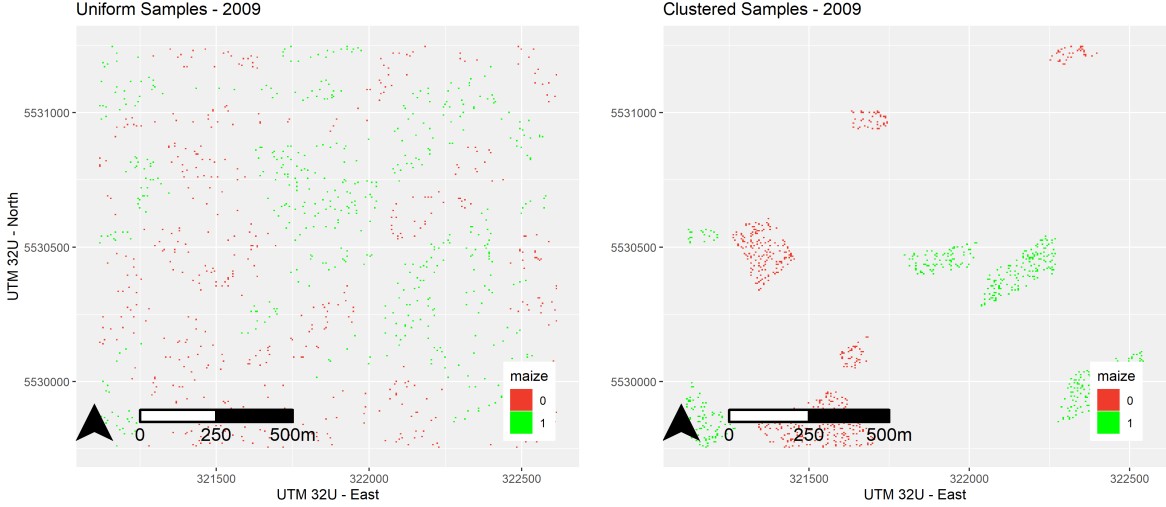

**Figure 4.** Uniform versus clustered sampling.

The second step of the model building process was the visual evaluation and optimization of the Medium-scale grid based sparse digitization of maize and non-maize polygons. Once the suitability of pixel based models was established in step 1, these were used to map uncertainty by visualizing probabilities that deviate substantially from 0 or 1. In cases where many of these pixels were clustered together, mistakes in digitizing the polygons were likely, and each of these cases was investigated closely.

Once a cleaned up reference dataset was finalized, a suitable model was applied to the predictor raster datasets. Since the predicted pixels are not useful enough on their own, statistics were aggregated on two levels. One single total area was computed for each year, in order to compare these predicted values to official data [27]. This is another important step to validate the prediction beyond the performances measured with the training data. Aside from performance measures purely based on the training data like kappa, error rates, sensitivity and specificity, independent official records are used to test model plausibility. In a last step, the maize area was aggregated spatially based on municipality

polygons [28], and relative values were calculated by dividing the maize area by the total agricultural area of each individual administrative unit.

## 3. Results

### 3.1. Small Scale Model Evaluation

Of the theoretical 9000 model computations, 8747 had meaningful outputs. This difference is a result of the clustered resampling because, in some resampling cases, the variogram could not be fit successfully, thus the Residual Kriging could not be performed. Figure 5 illustrates one random example of these runs. On this particular transect, two problematic areas with high residuals were identified. A closer look showed that these were small patches with less vegetation cover in the big field in the middle, which showed spectral properties closer to soil or grass.

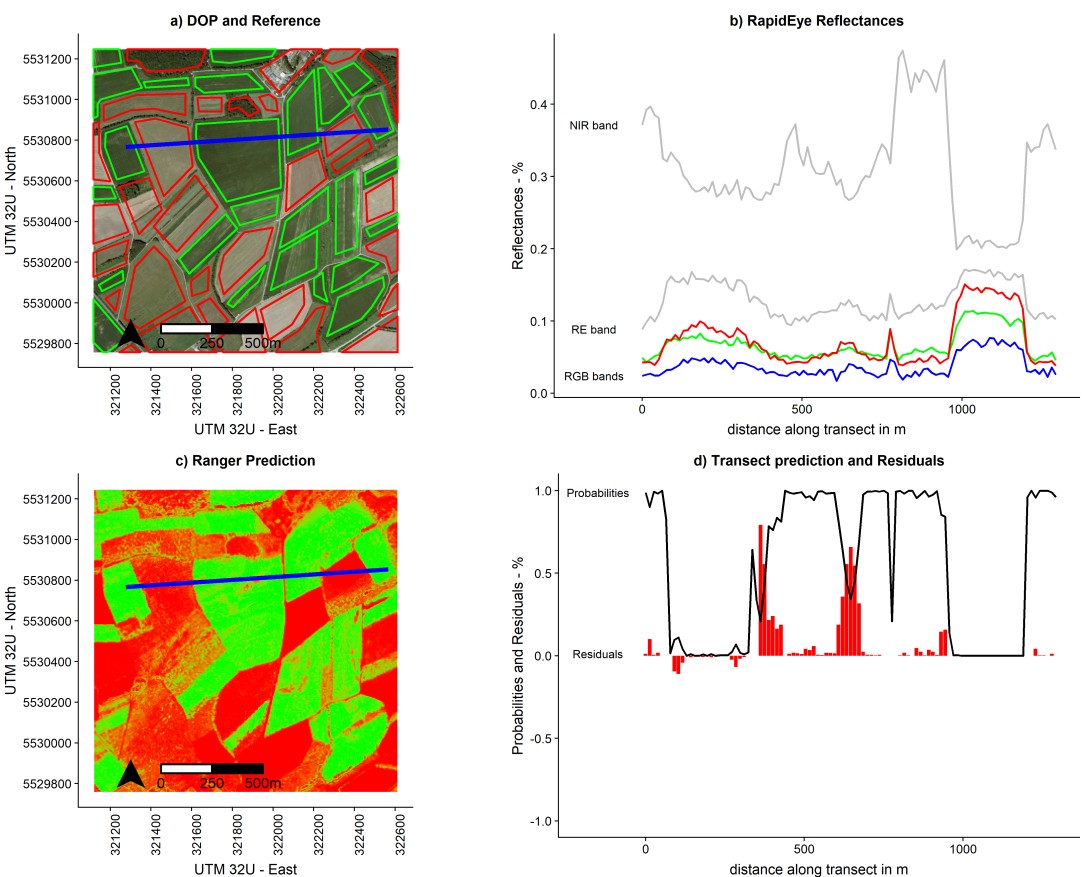

**Figure 5.** Example result of subset 2–2009–Random Forest (**a**) The high-resolution aerial image can be seen in the background, with the reference polygons for maize (green) and non-maize (red) on top. The blue line indicates the location of a transect, which is used to plot values in a 2D plot in (**b**). (**b**) The transect plot shows reflectance values for all five bands. In this plot, maize can be easily identified by looking at the peaks in the NIR band. (**c**) Probabilities of the random forest prediction are shown, with the green values being close to 1, and the red values being close to zero. (**d**) This 2D transect plot shows the probabilities and residuals along the transect.

All complete cases are summarized in Table 1, where the average kappa values for both years and sampling approaches were aggregated by the model. The kappa value is used to assess classification performance purely based on its popularity. Overall Accuracy was computed as well, with very similar results. To analyze the algorithms further, especially their impact on residual autocorrelation, empirical residual variograms were computed for all model residuals. A spherical model was then fitted to all empirical variograms, and the parameters were aggregated based on model type and sampling

approach for both years. While most of the fitted variograms showed no substantial nugget effect, the partial sill can be used to describe the impact of proximity on residual probabilities (see Table 2).

**Table 1.** Average kappa of small scale model runs.

| Approach | Uniform—2009 | Clustered—2009 | Uniform—2016 | Clustered—2016 |
|---|---|---|---|---|
| glm | 0.865 | 0.760 | 0.781 | 0.631 |
| glm + blur | 0.906 | 0.791 | 0.846 | 0.668 |
| glm + sk | 0.941 | 0.765 | 0.918 | 0.637 |
| ranger | 0.930 | 0.766 | 0.897 | 0.65 |
| ranger + blur | 0.958 | 0.784 | 0.937 | 0.668 |
| ranger + sk | 0.947 | 0.769 | 0.923 | 0.645 |
| svm | 0.931 | 0.769 | 0.893 | 0.678 |
| svm + blur | 0.949 | 0.784 | 0.925 | 0.694 |
| svm + sk | 0.964 | 0.779 | 0.949 | 0.666 |
| average pure | 0.909 | 0.765 | 0.857 | 0.653 |
| average blur | 0.938 | 0.786 | 0.902 | 0.677 |
| average sk | 0.951 | 0.771 | 0.930 | 0.650 |
| average glm | 0.904 | 0.772 | 0.848 | 0.645 |
| average ranger | 0.945 | 0.773 | 0.919 | 0.655 |
| average svm | 0.948 | 0.777 | 0.922 | 0.679 |
| average total | 0.932 | 0.774 | 0.896 | 0.660 |

**Table 2.** Averaged partial sills—fitted spherical variogram.

| Approach | Uniform—2009 | Clustered—2009 | Uniform—2016 | Clustered—2016 |
|---|---|---|---|---|
| glm | 0.0290 | 0.0642 | 0.0433 | 0.0865 |
| glm + blur | 0.0276 | 0.0617 | 0.0387 | 0.0821 |
| glm + sk | 0.0088 | 0.0639 | 0.0124 | 0.0872 |
| ranger | 0.0152 | 0.0613 | 0.0168 | 0.0753 |
| ranger + blur | 0.0148 | 0.0599 | 0.0165 | 0.0724 |
| ranger + sk | 0.0092 | 0.0605 | 0.0100 | 0.0763 |
| svm | 0.0168 | 0.0673 | 0.0186 | 0.0787 |
| svm + blur | 0.0163 | 0.0657 | 0.0172 | 0.0754 |
| svm + sk | 0.0052 | 0.0660 | 0.0071 | 0.0824 |

The kappa value is ranging from $-1$ to 1, and is intended to describe the performance of the prediction, with higher values indicating better performance. The partial sill, defined as the difference between the minimum and maximum variance of the fitted semivariogram, is intended to measure the residual autocorrelation of the prediction. It describes the difference in variability between two points that are close, and two points that are far away. Since we aimed to minimize spatial autocorrelation, lower values were preferred. As a first observation, all kappa values were reasonably high, ranging from 0.631 to 0.964. Furthermore, all partial sills were reasonably low, ranging from 0.0052 to 0.0872, which means that the difference in variance between points that are nearby and points that are far away is never above 0.1. There was also a significant difference between the two years, with the classification of the 2016 dataset having consistently lower kappa values and higher partial sills. RF and SVM classification performed substantially better than the GLM.

### 3.2. Visual Evaluation and Optimization of the Medium-Scale Grid Based Sparse Digitization of Maize and Non-Maize Polygons

The small scale model evaluation has shown that both SVM and RF classification were suitable to perform the necessary binary classification of maize and non-maize pixels. Since the RF approach does not rely on extensive hyperparameter tuning as much as the SVM, it is used in the further process.

In the next step, classification was performed without the Gaussian Blur on the dataset, with all bands and elevation as predictors of maize. Elevation was used as a proxy of the climatic conditions, which change slightly from north to south (see Section 2.1). Afterwards, errors in the training polygons were visualized as red pixels in a raster format, and as a circle in the center of each polygon with the size indicating the number of error pixels in each polygon. The size of the circles was used to get a good indicator of areas with critical error levels from the large scale view of the entire map, while the red pixels can then be used to get a detailed view of the spatial distribution in each digitized polygon. Once a problematic polygon is identified, the field ID can be used to track the polygon in the reference datasets and make adjustments if necessary.

In the example of Figure 6, fields with the number 39, 565 and 36 appeared to be problematic. The misclassifications in 39 and 36, maize pixels that are classified as non-maize, were vague in shape, and upon closer look were just maize pixels which have some bare soil showing because of worse growth conditions. These pixels were fine to be digitized that way because they were part of the inherent heterogeneity of the crop. The errors in field 565, however, non-maize pixels that were classified as maize, had a clear edge, which seemed to be part of a field to the north of the polygon. Upon closer look, it was clear that these pixels were actually maize pixels, and that polygon 565 was too large. With over 2500 digitized polygons, mistakes like this were quite common, but they could be easily identified and corrected with this approach.

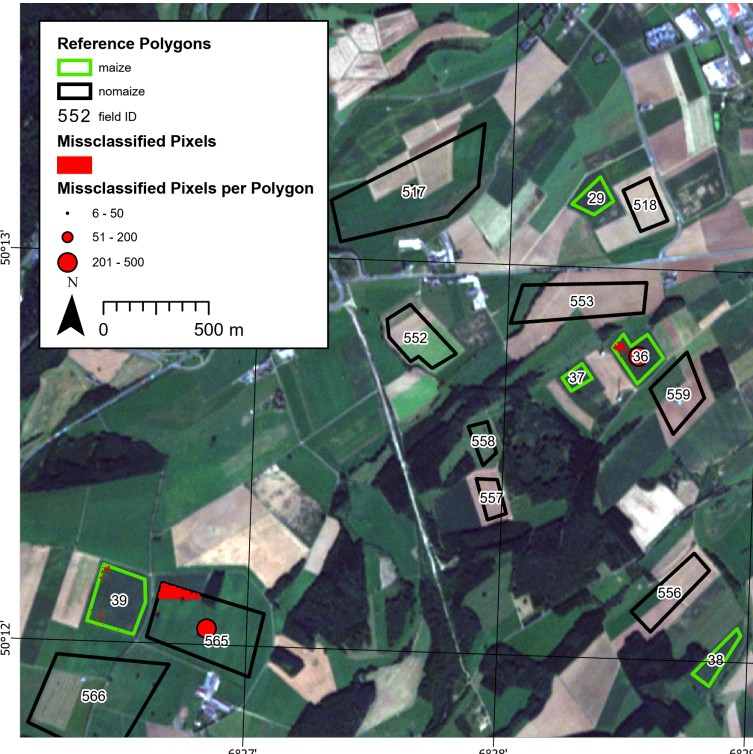

**Figure 6.** Digitization errors. The green and black polygons represent the digitized reference areas. The red circles indicate the number of misclassifications for any given polygon, while red pixels illustrate the location and spatial pattern of these misclassifications.

## 3.3. Pixel Based Mapping of Maize for the Entire Study Area

After the modeling approach was established, and the reference dataset was reevaluated, the RF classification and Gaussian Blur was applied to the entire region. The RF hyperparameters of mtry (number of variables used in a tree) and ntree (number of trees) were tested but had very little influence on model performance. Roughly 35 million pixels were classified, amounting to a total of 87,406 hectares of agricultural area. The performances are reported below in Table 3. The error rates

are Out-Of-Bag (OOB) meaning that these are the fractions of trees that were not trained with a given pixel, and predicted it incorrectly. Error rates and Kappa values were consistently good, while the classification in 2009 performed better than 2016, which was already a pattern in Section 3.1. Specificity was approximately 1, which means that there are very little false positives. Sensitivity, on the other hand, was quite a bit lower, which means that a lot of maize pixels went undetected. The total area of all maize pixels was then compared to official records [27], which showed an underestimation of 20% in 2009 and 27% in 2016. This discrepancy was substantially larger than the validation of the reference data would suggest.

**Table 3.** Overall classification results.

| Year | Error (OOB) | Kappa | Spec. | Sens. | Area (ha) Estimated | Area (ha) Official | Rel. Diff. (%) | Abs. Diff. (ha) |
|------|-------------|-------|-------|-------|---------------------|--------------------|----------------|-----------------|
| 2009 | 0.005 | 0.97 | 1 | 0.97 | 7305 | 9147 | 20 | 1842 |
| 2016 | 0.013 | 0.93 | 1 | 0.91 | 8447 | 11496 | 27 | 3049 |

In a final step, the prediction rasters were aggregated based on municipality polygons, and relative percentages were calculated based on the ratio of maize area and agricultural area for any given municipality. The final result (see Figure 7) was then mapped as the changes in relative percentages of maize area for a given municipality. This way, specific key regions could be separated from most areas with very little relative changes. Furthermore, a closer look at the prediction raster gave a detailed picture of the development in certain regions—for instance, the red area south of Prüm, where relative changes were larger than 20%, and many larger maize fields defined the landscape in 2016.

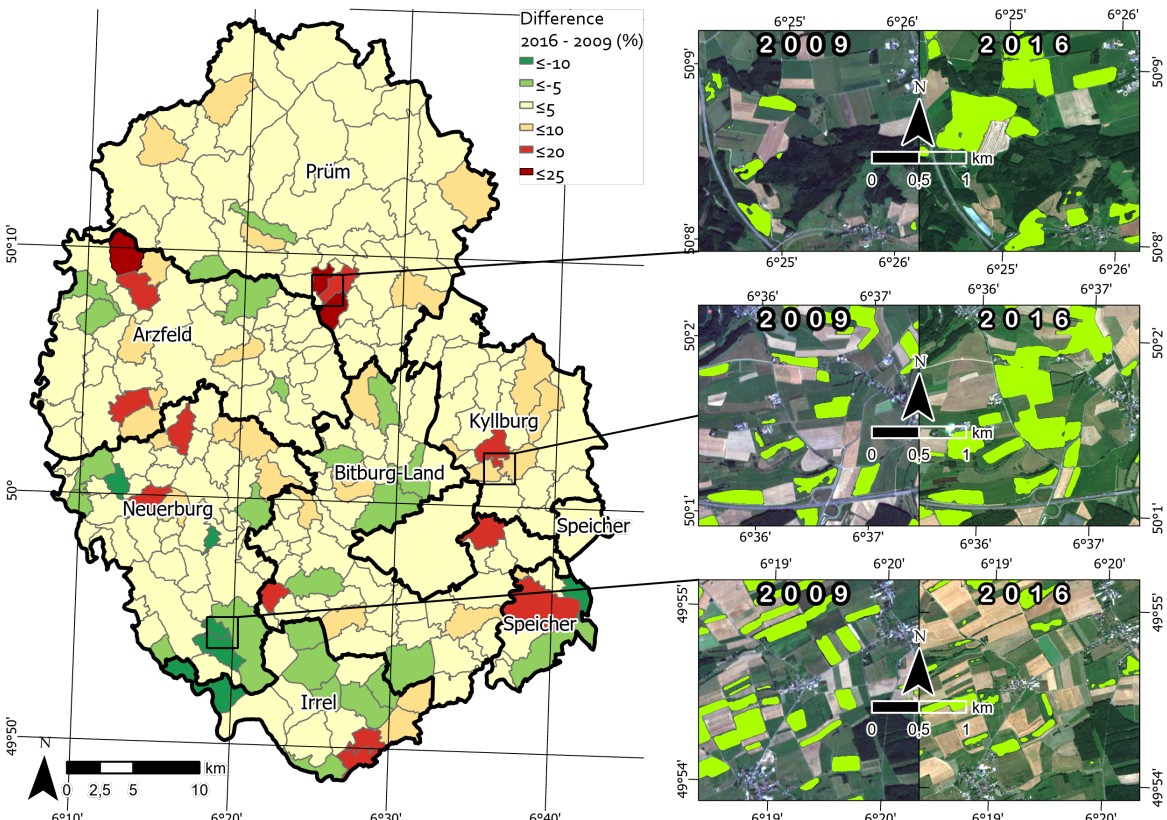

**Figure 7.** Changes in maize abundance. The six smaller images are true color RapidEye composites, with predicted maize fields in green. The map of the study area shows differences in the percentage of maize fields of the total agricultural area.

## 4. Discussion

### 4.1. Small Scale Model Evaluation

**Key finding 1**: *RF and SVM perform similarly well, and comparatively better than GLM.*

Both SVM and RF classification performed substantially better than the GLM, with Kappa values usually well above 0.9 in the case of uniform sampling distribution. This aligned well with the findings of methodical classification reviews, as described in Section 1.2, which mostly pointed out the strength of machine learning classifiers in comparison to statistical tools, which usually rely on assumptions about the distributions of the covariates. RF and SVM do not rely on these kind of assumptions, and are able to optimize nonlinear separation problems. While the maize class was mostly homogeneous (see also Section 2.3), the non-maize class consisted of pixels as diverse as bare soil, which is very dark in the infrared, and vegetation pixels like pastureland, which is quite bright in the infrared. This leads to possible bimodal distributions for some of the covariates, which is not a problem for machine learning algorithms.

The choice made towards RF classification done later in the study was mostly practical because of the more complex hyperparameter tuning SVMs require. While RFs also have hyperparameters, they are not as critical, optimizing them is rather a trade-off between performance and runtime, as opposed to over and underfitting the training data. This is usually the case when optimizing the cost parameter in Support Vector Classification. It is quite possible that a finely tuned SVM classification might have led to better results, as would have other potentially more intricate machine learning algorithms. The conclusion is not that RF classification is the best possible approach, but rather that it is highly adequate—in particular, since it is questionable if a perfect classification is possible or even desirable due to natural noise in the training data (see also Section 2.3).

**Key finding 2**: *Regression Kriging increases performance and decreases spatial autocorrelation when samples are uniformly distributed, but not when they are clustered.*

The RK approach worked as intended when samples were uniformly distributed, and was able to produce the highest kappa values, as well as very flat variogram models with the lowest overall sills. When clustered sampling was simulated, Kappa values did not improve as much and were even worse in comparison to the pure pixel based classifier. The same can be said for the estimated sills of the variogram models, where little to no improvement could be measured. That observation in itself was no surprise, since clustering of samples has been indicated as a problem by [14] as well.

Figure 8 shows why this is the case. Simple Kriging heavily alters the results in the direction of high residuals that are nearby, and does almost nothing for uncertain pixels if there are no high residuals around.

The results also showed that the classification performance was generally a lot worse in the clustered case, meaning that residuals are much higher. This is because five polygons were chosen in each run and each class, some of which were potentially very small or not representative for other reasons. Therefore, while in the clustered and uniform case, the same number of training pixels was drawn each run; the overall variability, especially of the maize training data, was potentially distinctly lower.

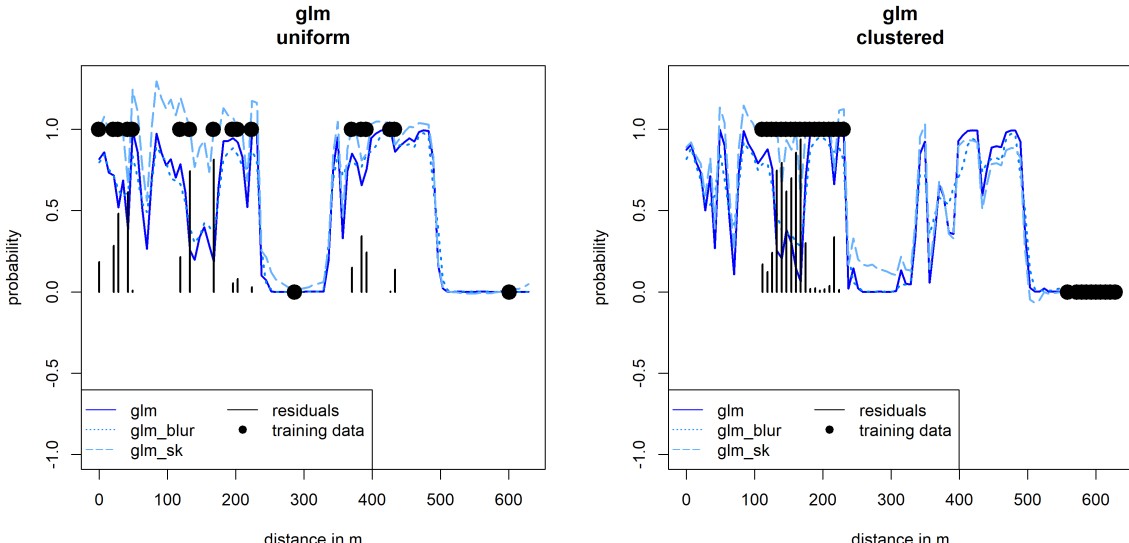

**Figure 8.** Transect of maize probabilities. In both plots, the solid blue lines indicate the predicted probabilities along the same transect. Training pixels are visualized with the black dots, uniform on the left, clustered on the right. The light blue line with the longer dashes shows the probabilities adjusted with the Simple Kriging. The difference is very large near large residuals. When no residuals are present, or when residuals are close to zero, Regression Kriging and glm probabilities are very similar. Gaussian blur probabilities, indicated by the lines with shorter dashes, work independently.

**Key finding 3**: *Gaussian Blur increases performance and decreases spatial autocorrelation when samples are uniformly distributed and when they are clustered.*

Clustered samples are not as problematic for the Gaussian Blur approach, since all neighboring pixels influence the probability equally depending on their distance, and not on whether they contain high residuals, or residuals at all. Only slight adjustments were made, as is shown in Figure 8. This mostly improved the classification result by removing noise in the training data, and only slightly decreased sills but for both clustered and uniform samples. It is likely that the decreased sills were a result of overall lower residuals, and that the Gaussian Blur does not lower residual autocorrelation at all in itself. The two issues of model performance and residual autocorrelation were hard to isolate, since it was not possible to change one without the other.

### 4.2. Pixel Based Mapping of Maize for the Entire Study Area

**Key finding 1**: *Classification of both years performs quite well, but 2009 is better than 2016.*

Maize is generally easy to classify, since most agricultural areas in the region were either bare soil, harvested cereal fields and pastures or still growing maize in late august. Difficulties arose when drought stress severely influenced pigments and vegetation cover in maize. Analysis of red and near infrared reflectances in both years, as shown in Figure 9, suggested that the training pixels of maize in 2016 contained substantially more soil and less chlorophyll content, since it is spectrally more heterogeneous. The figure also illustrates the distribution of false negatives and false positives of the out of bag predictions. It shows that, in both years, false negatives and false positives look surprisingly similar, with no real indication as to why model performance was worse in 2016.

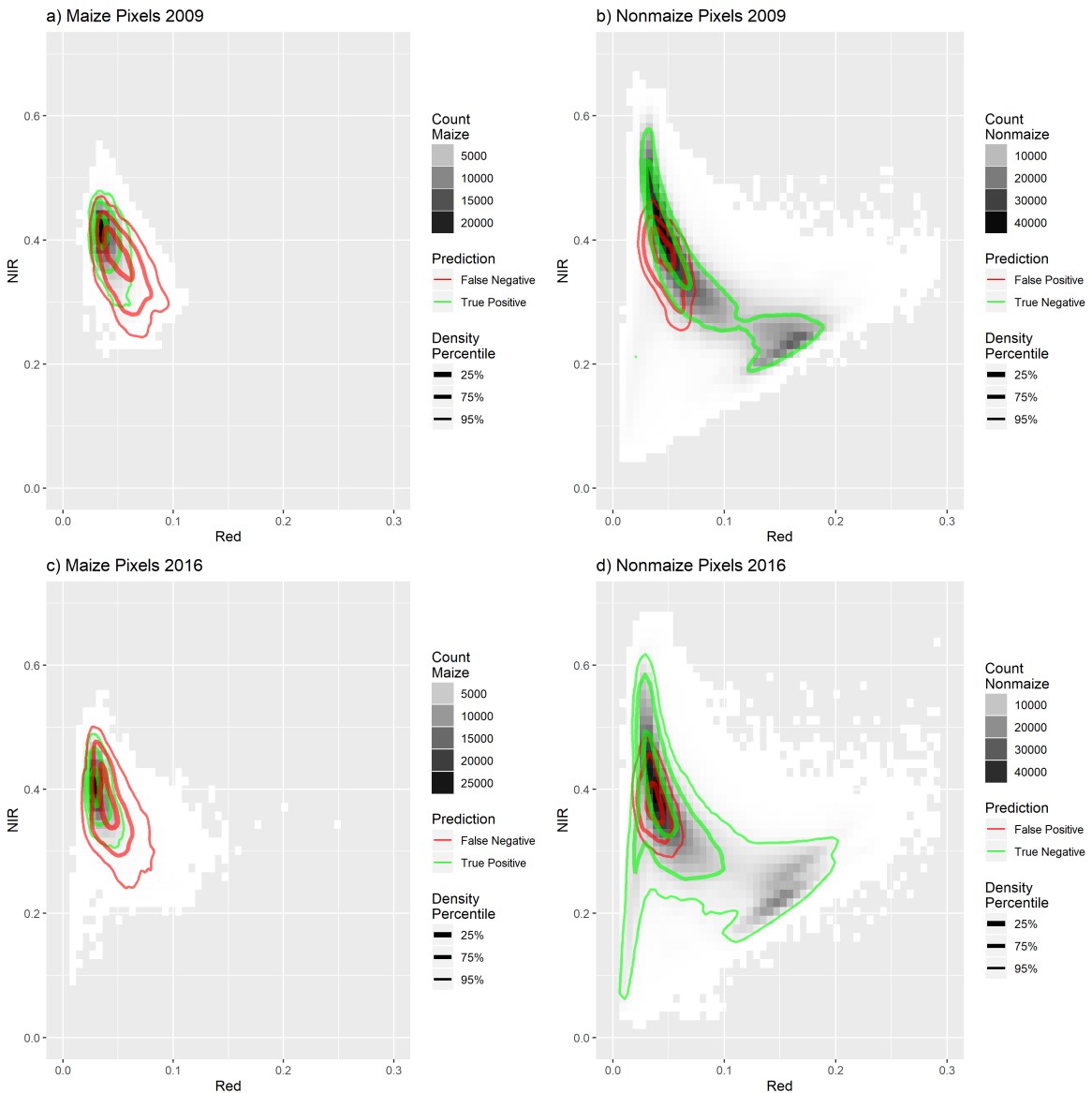

**Figure 9.** Spectral comparison of maize pixels 2009–2016.

This issue is ultimately impossible to deal with while using optical remote sensing alone, since it relies on the pigments of vegetation cover. Even if algorithms were applied to the satellite imagery to detect inhomogeneities in the spectra, excluding them would exclude a potentially large amount of total maize area in dry areas.

**Key finding 2**: *Disparity in estimated area and statistical records is substantially larger than validated modeled performances suggest.*

In Section 3.3, the estimated maize abundances were validated in two ways by comparing them to the reference, and by comparing them to the statistical record. Since the difference between estimation and reference was by far smaller than the difference between the estimation and the statistical record, there must be additional inconsistencies to consider. The disparity between the estimation necessarily consisted of two parts, error of the classification and error of the statistical data. The potential sources for errors in the classification were discussed above. The exact methodology of statistical records is unknown, but relies on farmers self-reporting their cultivation types based on the polygons of their fields. However, as mentioned in Section 2.3, some fields were missing a substantial amount of vegetation cover due to drought and other crop failures, so there naturally was a difference between

area with active maize vegetation cover versus area declared as maize by the farmer. This difference is very hard to determine, since the statistical record is not transparent and the data that it's based on is unavailable to researchers and the public. However, based on what is known about the quality of our estimation, it is likely that that the classification was closer to the truth than the statistical record.

## 5. Conclusions and Outlook

This study has shown that optical remote sensing is an adequate tool to monitor maize abundance in a spatially explicit way. Reference data could be generated reliably and efficiently by digitizing digital orthophotos. Optical remote sensing imagery could then be used to predict maize fields for any given pixel. To take the spatial relationships of all these pixels into account, we used two different approaches to incorporate spatial autocorrelation into our model. Regression Kriging, while very capable in theory, is not as beneficial when reference data are highly clustered. It is also technically not feasible when the sample count is very high, since it scales cubically. Gaussian Blur can help with residual noise, and scales only linearly, but is ultimately no systematic solution to the problem of residual autocorrelation. While we were not able to present a perfect way to deal with residual autocorrelation, we still underlined the importance of thorough residual analysis, since spatial classification has reliability indicators beyond overall accuracy measures or kappa values. The performance of machine learning classifiers on independent observations is well understood, but how to incorporate spatial relationships is still up for discussion. Combining pixels to objects and treating them as discrete entities might be a direct solution to the problem, known as Object Based Classification. It relies on segmentation techniques and high spatial resolution since homogeneous pixels are grouped together and treated as objects, while their aggregated attributes are then used with traditional classifiers like SVMs and Neural Networks [29]. Alternatively, computer vision approaches like Convolutional Neural Networks can be used to detect objects in remote sensing imagery, by computing additional kernel based feature layers [30].

The main technical limitation of this approach was the reliance on the reflectance of sunlight on leaf pigments. Firstly, this means there is simply no data when clouds cover the area of interests, which severely limits the availability of imagery, and makes it impossible to produce a time series for larger areas. Secondly, this makes it hard to include mostly brown or yellow canopies that are under severe drought stress. One way to deal with this is to use active microwave remote sensing with the combination of synthetic aperture radar (SAR) with optical remote sensing techniques. Due to the wavelength ranges of SAR sensors, they are mostly unaffected by clouds and other atmospheric attributes that are problematic for optical remote sensing. This enables the composition of gapless time series, not only from year to year, but with many observation points within one year [31].

**Author Contributions:** Conceptualization, M.G. and T.R.; methodology, M.G.; software, M.G.; validation, M.G.; formal analysis, M.G.; investigation,M.G.; resources, M.G. and T.R.; data curation, M.G.; writing—original draft preparation, M.G.; writing—review and editing, M.G.; visualization, M.G.; supervision, T.U. and C.E.; project administration, T.U. and C.E.

**Funding:** This research received no external funding. Imagery was provided by Planet Labs, Inc. (San Francisco, CA, USA) under the Rapideye Science Archive (RESA) progam. Project ID: 00197.

**Acknowledgments:** We kindly thank Petra Seiffert, Planet Labs Germany and the German Aerospace Center for providing RapidEye satellite images. Additionally, we thank David Frantz for assistance with the radiometric correction process.

**Conflicts of Interest:** The authors declare no conflict of interest.

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
