# Peer review of "Remote Sensing Based Binary Classification of Maize. Dealing with Residual Autocorrelation in Sparse Sample Situations"

_remotesensing, doi:10.3390/rs11182172_

Round 1
Reviewer 1 Report
please see my comments in the attached PDF. Thanks.

Author Response
Dear Reviewer,
Thank you very much for reading my manuscript and taking the time to offer some constructive criticism. Please consider the following remarks to your comments. I highlighted all my revisions in the document, structured by reviewer (R1 – R3) and made some additional adjustments (M1+M2).
Reviewer 1
R1.1 Problems in current research and contribution of this paper should be clarified
The manuscript is rather application-driven and does not intend to solve major theoretical problems. Its niche is the careful consideration of spatial effects in remote sensing data, without losing the reliability that comes with already well established algorithms like SVM and the like. We tried to achieve this by combining said algorithms with one (computationally) complex, and one very simple geostatistical tool (RK and Gaussian Blur). To clarify this, I made some adjustments in the abstract, the objectives and in the conclusion. They are highlighted in the pdf as:
R1.1.1 Minor clarification in the abstract (Line 6)
R1.1.2 Minor clarification in the objectives (Line 83)
R1.1.3 Restructured Conclusion, focus on spatial aspects (Line 404)
R1.2 Literature Research is not sufficient
Spatial effects in pixel based modeling is not exactly a new problem, and quite a few different approaches exist. The problem is that it is still ignored in a lot of studies, which is something that is hard highlight in literature research. I added some sources acknowledging already existing approaches, and underlined the fact that in two relatively recent (2014 and 2016) review papers about pixel based classification algorithms the issue is completely ignored. If you know other recently published material, I will happily consider your suggestions. The highlighted section contains my revisions:
R1.2.1 Added sources to the introduction, main goal is to illustrate discrepancy of available approaches and usage (Line 72)
R1.3 Additional Questions
R1.3.1 Which methods are compared with your proposed approach? (Line 98)
To clarify which models I combined with which spatial approach, I added a small paragraph in the objectives
R1.3.2 Why you choose these methods for comparison? (Line 220)
I added an outline of my basic reasoning about the model choices in the modeling section. If you have any specific suggestions about specific models I am happy to consider them.
R1.3.3 What about the parameters? (Line 213 and 303)
I added some remarks about the hyperparameters. I did not really optimize them for the 9000 modeling runs, since runtimes would have exploded. For the final model run I did a gridsearch for mtry and ntree, and found changing the default parameters had very little influence on model performance.
R1.3.4 What about the improvement of your proposed approach and analysis?
I was able to show that Regression Kriging tackles the issue of spatial autocorrelated residuals directly, but does not work in sparse and clustered sample situations. Gaussian Blur however improves the results independently from the training sampling, but does not reduce residual autocorrelation as well. I tried to clarify the results of my research by restructuring the abstract and the conclusion (see R1.1.1, R1.1.3).
R1.3.5 Please make sure the resolution of RapidEye satellite is 5 meters/pixel (Line 126)
The GSD is indeed 6.5 m, but the pixels are resampled in the orthorectification process to 5m. I added a small remark in the text
R1.3.6 Would you please explain how you obtain the ground reference map for accuracy evaluation? (Line 150)
I elaborated a little bit on the digitization process in the reference chapter. We digitized polygons by looking ad google earth imagery from that time period. Since maize is very prominent, it was easily distinguishable.
R1.4 Contribution and Conclusion
To clarify your other points I restructured my conclusion quite a bit, and tried to clarify the contribution of my manuscript in the abstract and the objectives. See R1.1.1, R1.1.2 and R1.1.3
I hope that my revisions will help to clarify our specific approach a bit. If you have further questions or suggestions, I will be happy to implement them. Thank you for your time!
PS: Some additional minor revisions
M1 North Arrows and Scale bars are missing
M1.1 Added grid, north arrows and scale bars to figures 1, 4, 5, 6, 7
M2 Sensitivity and Specificity switched
M2.1 Changed values and text in 3.3
Reviewer 2 Report
The authors presented a study that explores the monitoring of maize cultivation based on remote sensing data. They show an appropriate methodology to deal with this investigation, while the approach shows that Support Vector Machine (SVM) classification and Random Forests (RF) are able to distinguish maize pixels reliably. The results are supported in the discussion section by the key-findings which provide clear information to the readers. I found this research is well organized and provides sufficient background information. However, I would suggest the authors include an analysis of the false positives generated by the classifiers. As that would give a more specific view about the problems of each classifier to generate true positive samples that contribute to performance improvement. Finally, some minor spelling errors should be checked.
Author Response
Dear Reviewer,
Thank you very much for reading my manuscript and taking the time to offer some constructive criticism. Please consider the following remarks to your comments. I highlighted all my revisions in the document, structured by reviewer (R1 – R3) and made some additional adjustments (M1+M2).
Reviewer 2
R2.1 Analysis of False Positives
I agree very much that the analysis of false positives would be interesting to add, especially under the assumption that the specificity is much worse than the sensitivity in the overall out of bag predictions, as I said in my submitted manuscript. Unfortunately, this was an error, the positive factor level in the confusionmatrix function was not specified, and therefore, sensitivity and specificity are switched (see also M2). Specificity is quite high, which means there are very little false positives, and sensitivity is comparatively low, which is probably due to impurities of the training data.
But the point is still interesting, why are sensitivity and specificity so different. Since the computation of the models for the small scale subsets takes about a week, there was no time to rerun all of them. The final run of the medium scale dataset for the entire region could be analyzed, and I added contours of the red/near infrared distributions of false/true positives/negatives for both years. The difference between both years is quite small, and I still don’t have an explanation for the gap between sensitivity and specificity. If you have further suggestions on how this could be analyzed, I will be happy to implement them. You will find my revision highlighted here:
R2.1.1 Added oob prediction distributions to f9 and small text changes (378)
Thank you again for your comments on our study. Looking forward to hear from you again!
Some additional revisions
M1 North Arrows and Scale bars are missing
M1.1 Added grid, north arrows and scale bars to figures 1, 4, 5, 6, 7
M2 Sensitivity and Specificity switched
M2.1 Changed values and text in 3.3
Reviewer 3 Report
This is a good application-driven paper. The authors gave a more detailed introduction to the system. I still prefer to accept it, despite the lack of technical and theoretical contributions. My minor issues are as follows:
1. The experimental section gives some comparison results, but the comparison methods do not seem to be cutting-edge ones.
2. Can you release all the data sets used in the paper, such as the link to download, so that the readers can follow your work.
3. I think the authors should explain why the Kappa coefficient is used as an evaluation indicator.
Author Response
Responses to Reviewers
Dear Reviewer,
Thank you very much for reading my manuscript and taking the time to offer some constructive criticism. Please consider the following remarks to your comments. I highlighted all my revisions in the document, structured by reviewer (R1 – R3) and made some additional adjustments (M1+M2).
Reviewer 3
R3.1 Methods do not seem to be cutting edge
The methods chosen in this manuscript were chosen to be well established, and understood. Therefore it was not our intension to choose methods in order to maximize a given performance measure, but rather to explore possibilities to add a spatial dimension to widely used classifiers. So while it is true that the algorithms used in this paper are not particularly elaborate, the algorithms were able to solve the classification problem adequately. I tried to clarify the reasoning behind my modeling choices in R1.3.2 (Line 220), and furthermore stressed my research goals in R1.1.1 (Line 6) R1.1.2 (Line 83) and R1.1.3 (Line 404).
R3.2 Data availability
We contacted the provider of the datasets (Planet) immediately, but sadly received no answer yet. I fully agree that publishing the datasets with the manuscript is the optimal way, but I cannot make them available at this point.
R3.3 Explain why the Kappa coefficient is used as an evaluation indicator.
The kappa coefficient was used in this study purely based on its widespread use. While I am aware of the discussion around it, we put a lot of effort into not relying on any single performance indicator at all, and analyzed variograms of residuals, and compared our results to official records. Please do not hesitate to contact us again, if you have specific methodical doubts about the usage of kappa here, or suggestions for more helpful performance measures. Meanwhile I clarified my decisions a bit with the following revisions:
R3.3.1 Added a few remarks in 2.5 (Line 245 )
R3.3.2 Added a few remarks in 3.1 (Line 261)
I hope that my revisions will help to clarify our specific approach a bit. If you have further questions or suggestions, I will be happy to implement them. Thank you for your time!
Some additional revisions
M1 North Arrows and Scale bars are missing
M1.1 Added grid, north arrows and scale bars to figures 1, 4, 5, 6, 7
M2 Sensitivity and Specificity switched
M2.1 Changed values and text in 3.3
Round 2
Reviewer 1 Report
please see my comments in the attached PDF.

Author Response
Dear Reviewer,
Thank you very much for positively reviewing my revisions. In addition, I thank you for giving some literature recommendations. I found them useful. You are correct that the advances in Deep Learning are somewhat disregarded in my paper. My reasoning behind this is that they are mostly applied as object detection algorithms, rather than pixel based classifiers. Therefore I added one of your references to the conclusion, rather than the introduction, where I only talk about the pixel based approach. In the introduction I’ve added the paper about the training sample refining.
Thanks again!
R1.1.1 Added literature about sample refinement (Line 73)
R1.1.2 Added literature about CNNs (Line 417)